# Alterations in Natural Killer Cells in Colorectal Cancer Patients with Stroma AReactive Invasion Front Areas (SARIFA)

**DOI:** 10.3390/cancers15030994

**Published:** 2023-02-03

**Authors:** Nic G. Reitsam, Bruno Märkl, Sebastian Dintner, Eva Sipos, Przemyslaw Grochowski, Bianca Grosser, Florian Sommer, Stefan Eser, Pia Nerlinger, Frank Jordan, Andreas Rank, Phillip Löhr, Johanna Waidhauser

**Affiliations:** 1Pathology, Faculty of Medicine, University of Augsburg, 86156 Augsburg, Germany; 2General and Visceral Surgery, Faculty of Medicine, University of Augsburg, 86156 Augsburg, Germany; 3Gastroenterology, Faculty of Medicine, University of Augsburg, 86156 Augsburg, Germany; 4Hematology and Oncology, Faculty of Medicine, University of Augsburg, 86156 Augsburg, Germany

**Keywords:** SARIFA, Stroma AReactive invasion front areas, NK cells, natural killer cells, CRC, colorectal cancer, flow cytometry, tumor microenvironment

## Abstract

**Simple Summary:**

Our group recently presented Stroma AReactive Invasion Front Areas (SARIFA) as an independent prognostic biomarker for a poorer outcome in different gastrointestinal cancers, including colon cancer. We hypothesized that the lack of desmoplastic and inflammatory reactions in these carcinomas is based on immunologic alterations. To elucidate this phenomenon, we characterized the peripheral blood lymphocyte subsets and some parts of the local immune responses in colorectal cancer (CRC) patients, in relation to their SARIFA status. Through this, we could observe significant differences between SARIFA-positive and SARIFA-negative CRCs, especially with respect to natural killer (NK) cells. Modifying immune responses by using immunotherapy is already present in clinical routines. Therefore, understanding the basis of SARIFA is crucial for therapeutic strategies, specifically exploiting this new biomarker in CRC patients. This study reveals NK cells as potentially key players in this context, and they should be further investigated.

**Abstract:**

Background: Recently, our group introduced Stroma AReactive Invasion Front Areas (SARIFA) as an independent prognostic predictor for a poorer outcome in colon cancer patients, which is probably based on immunologic alterations combined with a direct tumor-adipocyte interaction: the two together reflecting a distinct tumor biology. Considering it is already known that peripheral immune cells are altered in colorectal cancer (CRC) patients, this study aims to investigate the changes in lymphocyte subsets in SARIFA-positive cases and correlate these changes with the local immune response. Methods: Flow cytometry was performed to analyze B, T, and natural killer (NK) cells in the peripheral blood (PB) of 45 CRC patients. Consecutively, lymphocytes in PB, tumor-infiltrating lymphocytes (TILs), and CD56+ and CD57+ lymphocytes at the invasion front and the tumor center were compared between patients with SARIFA-positive and SARIFA-negative CRCs. Results: Whereas no differences could be observed regarding most PB lymphocyte populations as well as TILs, NK cells were dramatically reduced in the PB of SARIFA-positive cases. Moreover, CD56 and CD57 immunohistochemistry suggested SARIFA-status-dependent changes regarding NK cells and NK-like lymphocytes in the tumor microenvironment. Conclusion: This study proves that our newly introduced biomarker, SARIFA, comes along with distinct immunologic alterations, especially regarding NK cells.

## 1. Introduction

With more than 900,000 deaths annually in 2020, colorectal cancer (CRC) contributes extensively to the burden of diseases globally [1].

Even though the five-year survival rate of CRC patients is 90% when diagnosed at an early stage, this rate rapidly decreases to 13% for those diagnosed later and reaching stage IV [2].

There are multiple approaches for predicting the outcome of CRC patients, in addition to the initial stage of diagnosis. One of them, the Immunoscore [3], which considers the amount of CD8+ and CD3+ lymphocytes at both the invasive front and central tumor parts of CRCs, has been extensively validated as a prognostic biomarker for CRC [4]. Although focusing on CD8+ and CD3+ cells in the context of IO (immune-oncology) therapy seems logical and useful, it is well known that the immune contexture of a tumor comprises many other cell types beyond CD8+ and CD3+ populations, which have potential prognostic and even predictive relevance too [5]. Therefore, many other immune cells, such as regulatory T cells (Tregs) [6], monocytes [7,8], and tumor-associated macrophages (TAMs) [9,10,11], have been shown to play a role in the progression of CRC and the outcomes of CRC patients.

Recently, our group [12] as well as other research groups [13,14] have been able to show that CRC patients exhibit alterations regarding their lymphocyte subsets in their peripheral blood. In addition, we demonstrated that these circulating lymphocytes, to some extent, even reflect the local antitumor immune response [15].

Additionally, our group recently introduced Stroma AReactive Invasion Front Areas (SARIFAs) as independent negative prognostic biomarkers in colon cancer and gastric cancer (GC) patients, which comes along with a direct tumor-promoting tumor-adipocyte interaction [16,17]. Moreover, we provided the first evidence for a specific metabolic interaction between adipocytes and tumor cells in the context of SARIFAs using digital spatial profiling (DSP) [17], which adds up to the already known important role of lipids and adipocytes in tumor progression [18]. Notably, Wulczyn et al. independently identified a morphologically very similar phenomenon (tumor cell clusters adjacent to fat tissue, here called “tumor adipose feature (TAF)”), using a deep learning approach [19], which was also associated with lower survival. Just recently, Foersch et al. similarly identified tumor cell/adipocyte co-localization as a so-far underappreciated morphological feature, which has an influence on the prognosis of CRC patients, by deploying Guided Grad-Cam images [20].

Compared to many other histomorphological or molecular features of CRCs, which have been established to guide therapeutic decisions, such as tumor budding [21], tumor deposits [22], consensus molecular subtypes [23], microsatellite status [24], and EGFR- or BRAF-mutation status, SARIFA does not need any further expensive testing and provides an extremely low interobserver variability [16,17]; thus, it is easy to implement into the diagnostic workflow. Furthermore, it is already known that adipocytes located next to cancer cells can promote tumor growth – for example, by providing energy for the cancer cells [25].

Now considering, in addition to our own results, the numerous, experimental studies on the major role of adipocytes and lipids in cancer [18], it is very likely that our hematoxylin-eosin (H&E) based SARIFA classification represents the morphological correlate of a certain aggressive tumor biology. Moreover, we believe this specific tumor biology is characterized by not only metabolic but also immunologic alterations. Utilizing the analyses of a previous study [15], we investigated the composition of peripheral blood lymphocytes and tumor-infiltrating lymphocytes (TILs), in relation to the SARIFA status.

Thus, this study connects two research approaches that we have previously considered separately by (i) analyzing the composition of peripheral blood lymphocytes in healthy controls vs. SARIFA-positive vs. SARIFA-negative CRC cases via flow cytometry; (ii) correlating our findings in the peripheral blood with local immune processes specifically at the invasion front (IF, SARIFA, and non-SARIFA) of the primary tumors via conventional immunohistochemistry (IHC).

## 2. Materials and Methods

### 2.1. Patient Cohort, Trial Design, and Ethical Approval

Our study cohort consisted of 45 patients diagnosed with CRC and treated by surgery, in addition to 27 healthy controls (healthy controls’ age; median: 55 (range: 49–61); healthy controls vs. SARIFA-negative, *p* = 0.038; healthy controls vs. SARIFA-positive, not significant at *p* < 0.05; healthy controls: male/female-ratio: 13/14). The control group composed of healthy blood donors from the blood bank of University Hospital Augsburg (Bavaria, Germany), which had been aged-matched in the best possible method and were also partly used in one of our previous publications [12]. According to the national guidelines for blood donation, our healthy controls had been routinely tested to exclude any acute or chronic diseases. CRC patients’ characteristics are displayed in Table 1. Patient inclusion took place at the University Hospital Augsburg over a two-year period (between December 2018 and November 2020). Previous investigations on the same prospective CRC cohort have also been published by us [12,15]. None of the patients received neoadjuvant treatment. Patients with confounding factors with a direct influence on the immune system, such as acute/chronic infectious diseases, immunodeficiency (inherited or acquired), and autoimmune disorders were excluded. The diagnosis of secondary tumors was also an exclusion criterion. Although operative treatment was scheduled according to guidelines for tumor stages UICC I–III, a small number of patients with a so-far unknown UICC stage IV prior to surgery were also included in the study. Our study was approved by the ethical committee of Ludwig Maximilian University of Munich (reference: Project number 18-726). It was also performed in accordance with the Declaration of Helsinki. Prior to inclusion, all patients consented to study enrollment by written informed consent.

### 2.2. Definition of SARIFA

SARIFA status was evaluated based on all available tumor slide sections, including the sections with the greatest depth of invasion. SARIFA status can be easily assessed using H&E stains. SARIFA was defined—in line with our previous publication regarding SARIFA in colon cancer [16]—as an area at the tumor invasion front (IF), in which at least one tumor gland or a group of at least five tumor cells are directly adjacent to adipocytes without interjacent inflammatory infiltrate, or without a desmoplastic stromal reaction (Figure 1). The stromal reaction can present as collagen formation, reactive histiocytes, or fibroblastic proliferation. Even if only a single SARIFA were present, for example, a single tumor gland surrounded directly by adipose tissue, the case was classified as SARIFA-positive [16,17]. The first author (N.G.R.) classified all tumors and was blind to the clinical data, clinical course, and other characteristics. In the rare cases where SARIFA status was not immediately obvious, the second author (B.M.), a board-certified pathologist, was consulted and a consensus diagnosis was made using a double-headed microscope.

### 2.3. Analysis of Lymphocyte Subsets via Flow Cytometry

Blood samples were taken prior to surgery and flow cytometry was performed within 24 h at our local laboratory using an FC 500 flow cytometer from Beckman Coulter (Brea, CA, USA). The gating strategy was used as previously published by our study group [12,15,26,27,28] and is displayed in detail in Appendix A. For cell staining, antibodies from Beckman Coulter (Brea, CA, USA) and Biolegend (San Diego, CA, USA) were applied. Absolute values of lymphocytes were calculated using leukocyte counts measured with Stem-Count (Stem-Kit, Beckman Coulter). Initial lymphocyte values were reported as percentages. Detailed information regarding the used antibodies and gating strategy is displayed in Appendix A. In brief, lymphocytes were identified by using forward and side scatter. B lymphocytes were defined by CD19 positivity and further subdivided into naïve, class-switched memory, non-class-switched memory, and transitional B cells. NK-like T cells were primarily identified (CD3+ CD56+) [29,30]. NK lymphocytes, which were of special interest in this study, were detected as CD56+ cells and subdivided into three functional NK subsets, namely CD56+ CD16+, CD56dim CD16bright, and CD56bright CD16dim. T cells were identified by CD4 or CD8 positivity, and both were further subdivided into memory, naïve, central memory, effector memory, and effector memory RA (herein called “EMRA”) cells. To further analyze CD8+ cells, they were divided into early, intermediate, late, or exhausted and terminal effector cells. CD4+ T helper cells were classified into Th1, Th2, and Th17/22. Moreover, CD4+ and CD8+ were subdivided into activated and regulatory cells.

### 2.4. Immunohistochemistry

All IHC was performed on representative 2 µm slides on the Leica Bond RX automated staining system (Leica, Wetzlar, Germany), following the automated standard IHC protocol optimized for use on this platform. For each SARIFA-positive and SARIFA-negative case, 2-µm thick, whole-slide, formalin-fixed paraffin-embedded (FFPE) sections with the greatest depth of tumor invasion were used. FFPE sections were deparaffinized in the instrument, which was followed by epitope retrieval for 20 min at 95 °C in EDTA and peroxidase block for 5 min at 25 °C. The slides were incubated with antibodies against CD56 (MRQ-62 rabbit monoclonal antibody, 1:200, Cell Marque, Rocklin, CA, USA) and CD57 (NK-1 mouse monoclonal antibody, 1:100, Cell Marque, Rocklin, CA, USA), diluted in Dako’s antibody diluting solution (Dako, DM830) for 32 min at 42 °C. Chromogen detection and hematoxylin counterstaining were performed using a bond polymer refine detection kit (Leica, Cat. No.: DS9800).

To assess the number of NK cells and NK-like lymphocytes at the tumor center (TC) as well as at the SARIFAs and non-SARIFAs at the IF, CD56+, and CD57+ lymphocytes were counted manually. Even though CD56 is expressed in other immune cells such as dendritic cells, it is considered the characteristic phenotypic marker of NK cells [31]. CD57 served as a marker for a terminally differentiated, yet also highly cytotoxic NK cell subpopulation [32]. To reflect heterogeneity, three areas with the highest positive lymphocyte density per case (SARIFA-negative CRC: 3× TC, 3× non-SARIFA; SARIFA-positive CRC: 3× TC, 3× SARIFA, 3× non-SARIFA) were considered. Here, it is important to note that even within SARIFA-positive CRCs, there can be other invasion front areas that are not SARIFAs (Figure 1E), which were also investigated. Representative slides of CD56 and CD57 IHC as well as H&E stains were digitized using 3D Histech Pannoramic Scan II (3D Histech, Budapest, Hungary).

Immunohistochemical staining with CD3 (2GV6 rabbit monoclonal antibody, 1:100, Roche Ventana Medical Systems, Tucson, AZ, USA) CD8 (144B mouse monoclonal antibody, ready to use, Cell Marque, Rocklin, CA, USA) antibodies was performed on 2 µm whole slide sections in an automated manner on a Ventana BenchMark ULTRA platform with an iVIEW DAB detection system (Roche, Mannheim, Germany), as has been described by our group previously [15]. For the assessment of TILs, we used a different automatic approach than has already been described, in detail, in our previous publication [15]. We used digital quantifier software (VMscope, Berlin, Germany) since CD3+/CD8+ TILs are far more numerous than NK/NK-like cells, and automatic cell counting is apparently faster and at least equally consistent [33,34]. Due to the different workflows, CD3 and CD8 IHC slides were digitized with a Philips Ultrafast Scanner (Philips Health Systems, Hamburg, Germany). Since the TILs were counted automatically in the TC areas and at the invasive margin with high T cell density, it is important to clarify that it was solely SARIFAs that were counted as IF within the SARIFA-positive CRCs. As mentioned earlier, this is because non-SARIFAs are present at the invasive margin, even within SARIFA-positive CRCs, as visualized in Figure 1E. Therefore, with respect to TILs, we only distinguished between SARIFA-positive and SARIFA-negative CRC cases and not specifically between the invasive margins containing SARIFAs with direct tumor-fat interactions and those without SARIFA.

### 2.5. dMMR/MSI Testing

For the detection of deficient mismatch repair (dMMR), IHC staining, which shows a high consistency with PCR-based MSI-testing [35], was performed using the purported two-stain method to initially only test for PMS2 and MSH6 [36,37]. For further identification of the MSI status, for example, in the case of unclear IHC results, molecular MSI testing was performed on normal DNA and tumor DNA by investigating five microsatellite markers via multiplex amplification. Both these methods and the materials used have been described in detail in our previous publication [38].

### 2.6. Next-Generation Sequencing

To further characterize our cohort on a molecular level, we performed next-generation sequencing (NGS). For DNA extractions from formalin-fixed, paraffin-embedded (FFPE) samples, a pathologist manually marked the tumor areas on an H&E-stained slide. The corresponding area on the slide, with at least 20% tumor cellularity, was macro dissected from 10 µm sections (2–3×). For the library preparation, the multiplex PCR-based AmpliSeq Illumina Focus Panel or AmpliSeq for Illumina Cancer Hotspot Panel v2 (Illumina, San Diego, CA, USA) was used. DNA was isolated with a Maxwell FFPE Plus DNA Kit (Promega, Madison, WI, USA) and RNA with a Maxwell RSC RNA FFPE Kit (Promega), according to the manufacturer’s recommendations. Concentrations were determined via a fluorometric method with Quantus (Promega). Consecutively, RNA was transcribed into cDNA via reverse transcriptase. The Focus Panel consisted of 269 (DNA) and 284 (RNA) primer pairs for the detection of hot-spot regions and full genes of 52 genes with a targeted size of 29 kb (DNA) and 26 kb (RNA). The Hotspot Panel consisted of 207 primer pairs for the detection of Hotspot regions in 50 genes with a targeted size of 22k kb (DNA). Our laboratory workflow for NGS and used materials have been described extensively in our previous publication [38]. Data analysis was performed using the Illumina Base Space application for DNA/RNA amplicons and the subsequent interpretation using the Variant Interpreter (Illumina), in which non-synonymous and non-polymorphic changes were filtered. Of the remaining mutations, those with an allele frequency below 5% were not considered. Mutations below this threshold were individually matched using Integrated Genome Viewer (IGV, Broad Institute) with the human reference genome hg19 to exclude sequence errors. Moreover, data were analyzed with the BaseSpace Knowledge Network and the variants were documented accordingly.

Analyzed genes with AmpliSeq Focus Panel (Illumina):Analysis of mutations in the following genes: AKT1, ALK, AR, BRAF, CCND1, CDK4, CDK6, CTNNB1, DDR2, EGFR, ERBB2, ERBB3, ERBB4, ESR1, FGFR1, FGFR2, FGFR3, FGFR4, GNA11, GNAQ, HRAS, IDH1, IDH2, JAK1, JAK2, JAK3, KIT, KRAS, MAP2K1, MAP2K2, MET, MTOR, MYC, MYCN, NRAS, PDGFRA, PIK3CA, RAF1, RET, ROS1, and SMO.Analysis of fusions in the following genes: ABL1, ALK, AKT3, AXL, BRAF, EGFR, ERBB2, ERG, ETV1, ETV4, ETV5, FGFR1, FGFR2, FGFR3, MET, NTRK1, NTRK2, NTRK3, PDGFRA, PPARG, RAF1, RET, and ROS1.

Analyzed genes with AmpliSeq for Illumina Cancer Hotspot Panel v2 (Illumina):

ABL1, EGFR, GNAS, KRAS, PTPN11, AKT1, ERBB2, GNAQ, MET, RB1, ALK, ERBB4, HNF1A, MLH1, RET, APC, EZH2, HRAS, MPL, SMAD4, ATM, FBXW7, IDH1, NOTCH1, SMARCB1, BRAF, FGFR1, JAK2, NPM1, SMO, CDH1, FGFR2, JAK3, NRAS, SRC, CDKN2A, FGFR3, IDH2, PDGFRA, STK11, CSF1R, FLT3, KDR, PIK3CA, TP53, CTNNB1, GNA11, KIT, PTEN, and VHL.

As different panels were used to characterize our cohort (some in routine diagnostics, some for research use only), we logically compared only the mutational status of genes that were well covered by both panels between our SARIFA-positive and SARIFA-negative CRC patients. Variant calling and interpretation were performed identically.

The overlapping genes (subsequently used for subgroup analysis) included the following:

ABL1, AKT1, ALK, BRAF, CTNNB1, EGFR, ERBB2, ERBB4, FGFR1, FGFR2, FGFR3, GNAQ, HRAS, IDH1, IDH2, JAK2, JAK3, KIT, KRAS, MET, NRAS, PDGFRA, PIK3CA, RET, and SMO.

### 2.7. Statistical Analysis

Descriptive analysis results were reported as median values and interquartile ranges. For hypothesis testing of the differences between relative frequencies, Fisher exact tests were applied. Continuous variables were compared by using the Wilcoxon rank-sum test. When comparing three groups (healthy controls, SARIFA-negative, and SARIFA-positive), we performed a pairwise Wilcoxon test with Benjamini-Hochberg correction and reported adjusted *p*-values. The *p*-values are reported as follows: * *p* < 0.05, ** *p* < 0.01, *** *p* < 0.001, **** *p* < 0.0001. Data were analyzed using either SPSS for Windows, version 24 (IBM, Armonk, NY, USA) or R, version 4.2.1 (R Foundation for Statistical Computing, Vienna, Austria). The figures were designed using R, version 4.2.1.

## 3. Results

### 3.1. Patient Characteristics and NGS Analysis

The clinical characteristics of our patient cohort are summarized in Table 1 (see Section 2.1.). A total of 15 (33.3%) out of 45 patients showed a SARIFA-positive CRC. SARIFA-positive CRC patients were significantly younger (*p* = 0.043) and predominantly men (*p* = 0.012). Regarding the UICC stage, tumor location, and microsatellite status, the two cohorts did not differ significantly.

To characterize our cohort molecularly beyond MSI and in more detail, NGS-based multigene testing was performed on 38 of 45 CRC specimens. By comparing the relative frequencies of the observed mutations in SARIFA-positive and SARIFA-negative CRCs, we could show that none of these mutations are over-represented in any of our subgroups. Hence, we proved, at least for our cohort, that SARIFA-positive and SARIFA-negative CRC are both heterogenous groups regarding their tumor genetics and did not differ significantly (Table 2).

### 3.2. Flow Cytometry-Based Analysis of Peripheral Blood Lymphocytes

Flow cytometry was performed to compare the numbers of peripheral blood lymphocytes and their subsets between SARIFA-negative and SARIFA-positive CRCs. To put our findings into context, a healthy control group was also examined. Flow cytometry revealed no differences between total lymphocytes, CD8+ and CD4+ T cells, and all their subsets, in addition to B cells and B cell subsets in SARIFA-positive versus SARIFA-negative patients. In contrast, NK cells were significantly reduced in SARIFA-positive patients (total numbers: 87/µL vs. 187/µL; *p* = 0.002). NK cell subsets also showed lower values in SARIFA-positive patients for CD56dim CD16bright NK cell subsets (6/µL vs. 14/µL; *p* = 0.004) and for CD56+ CD16+ NK cells (56/µL vs. 151/µL; *p* < 0.001) compared to SARIFA-negative CRC patients and healthy controls. Moreover, SARIFA-negative and SARIFA-positive CRC patients both showed significantly increased levels of HLA-DR+ CD4+ and CD8+ cells. The results of the flow cytometric analysis on the circulating lymphocytes are displayed in Figure 2 and Table 3. The differences between NK cells and NK cell subsets dependent on SARIFA status are specifically illustrated in Figure 3.

### 3.3. Immunohistochemical Analysis of Tumor-Infiltrating as well as CD56+ and CD57+ Lymphocytes

To correlate our findings in the peripheral blood with the immune cell infiltrate in the tumor center (TC) and specifically at the tumor invasion front (IF), we performed conventional immunohistochemistry. By counting the TILs automatically using CD3 and CD8 IHC, we did not observe differences regarding TILs between SARIFA-positive and SARIFA-negative CRCs (IF (not necessarily SARIFAs, see Section 2.4) CD3: *p* = 0.75, CD8: *p* = 0.23; TC CD3: *p* = 0.58, CD8: *p* = 0.26, Appendix A).

Considering our flow cytometry results indicated that there was a reduction in NK cells in the peripheral blood of SARIFA-positive CRC patients, we immunostained for CD56 as a surrogate marker for NK cells/NK-like lymphocytes and CD57 as a marker for a terminally differentiated, cytotoxic NK cell subpopulation. Here, we assessed the amount of CD56+ and CD57+ lymphocytes in the TC and at IF. In SARIFAs, we observed a statistically significant decrease of CD56+ as well as CD57+ lymphocytes (CD56 in Figure 4 and CD57 in Figure 5: CD56: *p* < 0.0001, CD57: *p* < 0.0001). CD56+ lymphocytes were decreased at SARIFAs even within SARIFA-positive CRCs, when comparing IFs (SARIFAs and non-SARIFAs) within SARIFA-positive cases (see Figure 1E, CD56: *p* < 0.01, CD57: *p* = 0.081, not significant at *p* < 0.05). As the invasive margin is, itself, heterogenous, there are non-SARIFAs even in CRCs that are classified as SARIFA-positive. This is depicted in Figure 1E and also described in detail in Section 2.4 (Methods: Immunohistochemistry). The results of the comparison of IFs, with and without SARIFA, in SARIFA-positive CRCs are displayed in Figure 4D and Figure 5D. In the TC, there was no difference between SARIFA-positive and SARIFA-negative CRCs (CD56: *p* = 0.15, CD57: *p* = 0.85).

## 4. Discussion

This is the first study to specifically focus on the immunologic abnormalities of SARIFA-positive CRCs. By comparing the peripheral lymphocyte status of SARIFA-positive and SARIFA-negative CRC cases via flow cytometric analysis, we could see that NK cells are heavily reduced in the peripheral blood of SARIFA-positive patients compared to healthy controls and SARIFA-negative patients. In line with our previous findings [12], both CRC subgroups presented with increased numbers of HLA-DR+ CD4+ and CD8+ peripheral blood lymphocytes compared to healthy controls, which points to an unspecific activation in the context of malignancy. Based on our findings regarding NK cells, we immunostained primary CRCs for CD56 and CD57. Our immunohistochemical studies suggest a significant reduction of NK cells and NK-like lymphocytes at SARIFAs. Interestingly, CD57+ lymphocytes, which are terminally differentiated senescent cells with a cytotoxic effector and memory-like functions [32], seem to be generally decreased at the IF of SARIFA-positive CRCs, and not solely at SARIFAs. Contrastingly, the number of CD56+ lymphocytes is pronouncedly decreased at SARIFAs. Hence, our study shows that circulating NK cells reflect, at least to a certain degree, the local antitumor response at SARIFAs. Using a different automatic counting approach, we could not detect any differences between SARIFA-positive and SARIFA-negative CRCs regarding CD3+/CD8+ TILs. As we have already demonstrated that macrophages—likely of M2 polarization—are upregulated at SARIFAs in GC [17], our data hint at another aspect of the immunosuppressive microenvironment at SARIFAs, which even reflects to some part in the peripheral blood. Our data suggest a pivotal role of NK cells in the context of SARIFA. There is already a considerable amount of data proving that low peripheral blood NK cell counts [39] as well as reduced levels of CD56+ [40,41], and CD57+ [42,43,44,45] tumor-infiltrating lymphocytes are associated with a poorer prognosis in CRC [46], which correlates with our results in SARIFA-positive cases. However, the data regarding tumor-infiltrating NK cells are not completely unequivocal, showing a clear trend toward an association between the reduction in the number of cells and adverse outcomes [46,47]. In this context, seminal work performed by Väyrynen et al. shows that NK cells and NK-like lymphocytes are spatially more closely related to tumor cells and that higher densities of these comparably rarer immune cell types in the CRC microenvironment are associated with lower cancer-specific mortality [48]. As NK cells are a crucial part of the innate antitumor immune response by performing cytotoxic functions, it seems biologically reasonable that low NK cell counts are associated with reduced antitumor activity. NK cells do not solely rely on MHC presentation and can even kill tumor cells that downregulate MHC as an immune evasive mechanism (in the absence of other inhibitory signals or soluble factors inhibiting NK-cell cytotoxicity) [49]. NK cells can be divided into functional subgroups with different antitumor activities [50,51]. Interestingly, our data point to a general reduction of NK cells, indicating a widespread involvement of NK cells in SARIFA-positive CRCs, as we found decreased levels of NK cells in general alongside the CD56dim CD16bright and CD56+ CD16+ subtypes. As CD56dim NK cells express higher levels of CD16, the FCgamma receptor III, as well as Ig-like NK receptors, CD56dim CD16bright NK cells are more cytotoxic [52]. Lower numbers of cytotoxic NK cells could causally explain the reduced antitumor activity in SARIFA-positive CRC patients, which could eventually contribute to a poorer prognosis.

Having already shown that peripheral blood lymphocytes reflect the local tumor immune response [15], we checked if this is also true for NK cells. Here, we observed that the downregulation of CD56+ lymphocytes in SARFA-positive CRCs is restricted to SARIFAs: areas at the invasion front, which are characterized by tumor-associated fat cells without desmoplastic or inflammatory reactions. It is already known that cancer-associated fat cells can contribute to an immunosuppressive tumor microenvironment [18,53,54] and that lipid accumulation can cause NK-cell dysfunction [55]. The same is also known for the abovementioned M2 macrophages [10,56]. Furthermore, two independent deep learning models highlighted the importance of tumor cell/adipocyte co-localization [19,20] and another the relevance of inflamed fat as a risk factor for lymph node metastasis in early CRCs [57]—both confirming the important role of adipose tissue in CRC.

As we did not observe any differences between SARIFA-positive and SARIFA-negative CRCs regarding the number of NK cells in the tumor center, our results further strengthen the outstanding relevance of the invasive border in CRC [58], where immune cell density is known to be higher than in the tumor center [59,60] and where our SARIFA-phenomenon is, by definition, located.

Using NGS-based multigene testing, this is the first study ever that provides a molecular characterization of CRCs, with regard to their SARIFA status; thus, we could show for our cohort that both SARIFA-positive and SARIFA-negative CRCs are molecularly heterogeneous and are seemingly not defined by certain specific driver mutations, or a predominant microsatellite status that could cause the immunologic differences, which is, for example, known for *KRAS* mutations [61,62,63]. However, to prove this, both a larger cohort and a larger NGS panel will be necessary for future studies. Even though we used two different panels, due to logistical reasons, it is well known that different NGS panels show a high concordance [64,65]. Logically, we only took the mutational status of the genes into account, which are well covered by both panels. Moreover, both panels are produced by the same manufacturer (Illumina) and used in our routine diagnostic workflow. Additionally, variant calling and interpretation were performed identically.

Our study provides further evidence that other parts of the immune system, beyond the extensively studied CD3+ and CD8+ cells, can be dysregulated in CRC subgroups. Furthermore, our study reinforces the already existing view that the spatial distribution of immune cells (tumor center vs. invasion front (SARIFA vs. non-SARIFA)) and the composition are highly relevant, which has already been shown by numerous studies [59,66,67,68,69,70]. To really understand the immunologic background of SARIFAs and the specific tumor-adipocyte interaction, further studies that use advanced and more spatially resolved techniques, such as digital spatial profiling (DSP) [71,72], or multiplex immunohistochemistry in combination with computational pathology methods, could further illuminate this tumor-adipocyte-immune cell crosstalk.

As SARIFA-positive carcinomas make up a special subgroup with a poorer prognosis [16,17], there is a particular need for novel therapeutic approaches. Besides potentially targeting the metabolic changes induced by tumor-promoting adipocytes, our results put immunotherapeutic approaches on the map of potential therapeutic strategies for SARIFA-positive CRCs. Whereas the composition and the predictive value of adaptive immune cells in the TME in CRC have been comprehensively analyzed by Galon et al. almost two decades ago [66] and considered a potential therapeutic target since then, immunotherapy approaches targeting innate immune cells such as macrophages, neutrophils, and specifically, NK cells have only recently become the focus of increased attention [73,74,75]. Therefore, the combination of better stratification of patients (e.g., into SARIFA-positive/negative or by using the Immunoscore or similar approaches, potentially even in combination) and an improved understanding of the therapeutic potential of NK cells could lead to new, personalized therapeutic targets in CRC. Considering that anti-PDL1-therapy is already known to unleash an NK cell antitumoral response [76,77,78,79,80,81], it would be highly instructional to observe whether classifying the SARIFA status could serve as a predictive biomarker for immune-oncological (IO) therapeutic approaches [82]. The latter would be especially beneficial because the SARIFA status can be determined reliably, quickly, and inexpensively on H&E stains, thereby making it very different from all the previous IO therapeutic biomarkers and a protocol that could be very quickly integrated into the clinical workflow. Further unraveling the mutational landscape of SARIFA-positive CRCs, for example, regarding HRR mutations, could lead to even more precise combination therapies [38,83,84,85], especially in the case of refractory disease or relapse. Based on our findings, SARIFA status could also be of potential predictive value when using cetuximab, a monoclonal antibody that targets the epidermal growth factor receptor (EGFR), as it is known that cetuximab functions via NK-cell mediated antibody-dependent cell-mediated cytotoxicity (ADCC) [40,86]. Interestingly, besides KRAS/NRAS wildtype status (40%/93.3% of SARIFA-positive patients in our cohort), which is already an approved biomarker for cetuximab therapy, CD56+ cell infiltration has already been linked to response to cetuximab therapy in CRC patients [40].

As one of the limitations of this study, it should be noted that the number of patients included (*n* = 45) is still relatively small. However, despite this, the differences regarding NK cell numbers were highly significant. Furthermore, besides just the number of NK cells, their functional state is also important for antitumoral immunity, which this study did partially investigate, albeit only with regard to NK cell subtypes. It is well known that different NK cell subtypes exert different functions [32,50,52]. To characterize NK cells even more functionally, we are planning primary NK cell cultures derived from peripheral blood mononuclear cells (PBMCs) and in vitro stimulation experiments comparing functional differences between NK cells from patients with SARIFA-positive and SARIFA-negative CRCs. In this context, analyzing the degranulation potential, lytic mediator content, IFγ secretion, expression of activating/inhibiting receptors, including checkpoint receptors, and killing activity (e.g., via performing killing assays using PMBC-derived NK cells and targeting colon cancer cell lines) of NK cells with regards to SARIFA in further studies would be extremely beneficial. In this study, we immunostained only CD56 and CD57, which served as surrogate markers for NK cells (CD56) and a highly cytotoxic NK cell subpopulation (CD57) [32], respectively. Although CD56 is not solely specific to NK cells and is also expressed in NK-like T cells [29,30], dendritic cells, and monocytes, CD56 is considered the archetypical NK cell marker [31] and has been repeatedly used as an immunohistochemical surrogate marker for NK cells in different cancer entities including CRCs [40,41,46,87,88,89]. To characterize the functional relevance/activity of NK cells at SARIFAs in more depth, further investigations such as DSP, multiplexing, single-cell RNA sequencing, or studying the protein expression of proteins related to the NK cell-mediated cytotoxicity pathway [90] could add further insights. To better understand the role of CD3+ and CD8+ lymphocytes and possibly also of tertiary lymphoid structures in relation to SARIFA, further studies, deploying computational pathology approaches [91], are required. In addition, our molecular characterization of the majority of included cases gives an initial, yet not fully conclusive insight into the tumor genetics of SARIFA-positive CRCs.

## 5. Conclusions

In conclusion, our study proves that our newly introduced biomarker SARIFA is associated with distinct immunologic alterations: (i) a dramatic reduction in NK cells in the peripheral blood of SARIFA-positive CRC patients and (ii) a (potential) reduction of NK cells and NK-like lymphocytes at SARIFAs in CRC. Even though we believe the interplay between tumor cells, adipocytes, and immune cells at SARIFAs is a highly complex process and NK cells in the peripheral blood or locally represent only one aspect, our study reveals NK cells as potentially key players in the context of SARIFA that could be, at least to some extent, causally relevant to a poorer prognosis for SARIFA-positive CRC patients, and additionally, could be potentially exploited by the use of IO therapy. In summary, this study can be regarded as a potential starting point for further studies to investigate the complex interplay between tumor cells, immune cells, and tumor-associated adipocytes at SARIFAs.

## Figures and Tables

**Figure 1 cancers-15-00994-f001:**
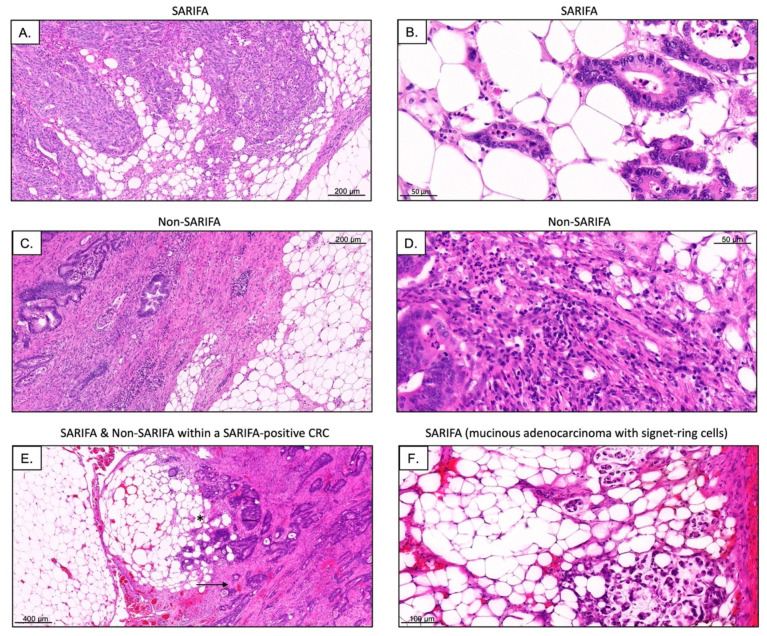
Representative H&E stains of SARIFAs as well as non-SARIFAs ((**A**,**C**); scale bar = 200 µm; (**B**,**D**); scale bar = 50 µm). Even in SARIFA-positive CRCs (SARIFA marked by *) some parts of the invasive margin are non-SARIFA (marked by an arrow; (**E**), scale bar = 400 µm). Additionally, rarer subtypes of CRC, such as mucinous adenocarcinoma with signet-ring cells, can show SARIFA ((**F**,) scale bar = 100 µm).

**Figure 2 cancers-15-00994-f002:**
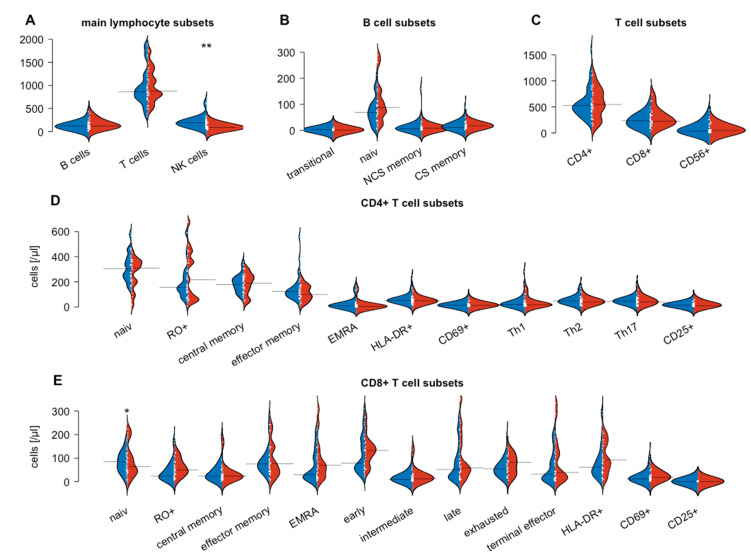
Comparison of lymphocyte subsets between SARIFA-positive (red) and SARIFA-negative (blue) CRC patients: Main lymphocyte subsets (**A**), B cell subsets (**B**), T cell subsets (**C**), CD4+ T cell subsets (**D**), CD8+ T cell subsets (**E**). Small white lines indicate individual data points; large black lines show the overall subset average of SARIFA-positive and SARIFA-negative CRC patients, respectively. * *p* < 0.05, ** *p* < 0.01.

**Figure 3 cancers-15-00994-f003:**
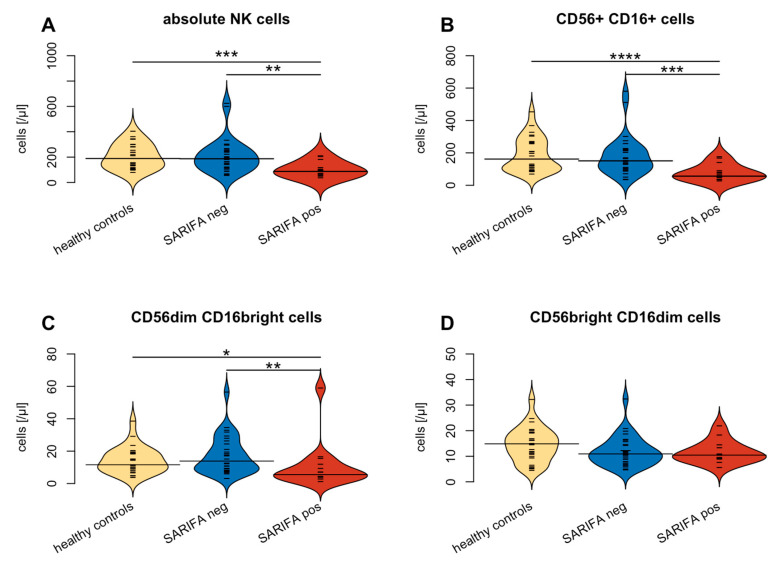
Comparison of NK cells in general (**A**) and CD56+ CD16+ subtype (**B**) as well as between CD56dim CD16bright (**C**) and CD56bright CD16dim (**D**) NK cell subtypes between healthy controls (yellow), SARIFA-pos(itive) (red) and SARIFA-neg(ative) (blue) CRC patients. Small white lines indicate individual data points; large black lines show the overall subset average of SARIFA-positive and SARIFA-negative CRC patients, respectively. * *p* < 0.05, ** *p* < 0.01, *** *p* < 0.001, **** *p* < 0.0001.

**Figure 4 cancers-15-00994-f004:**
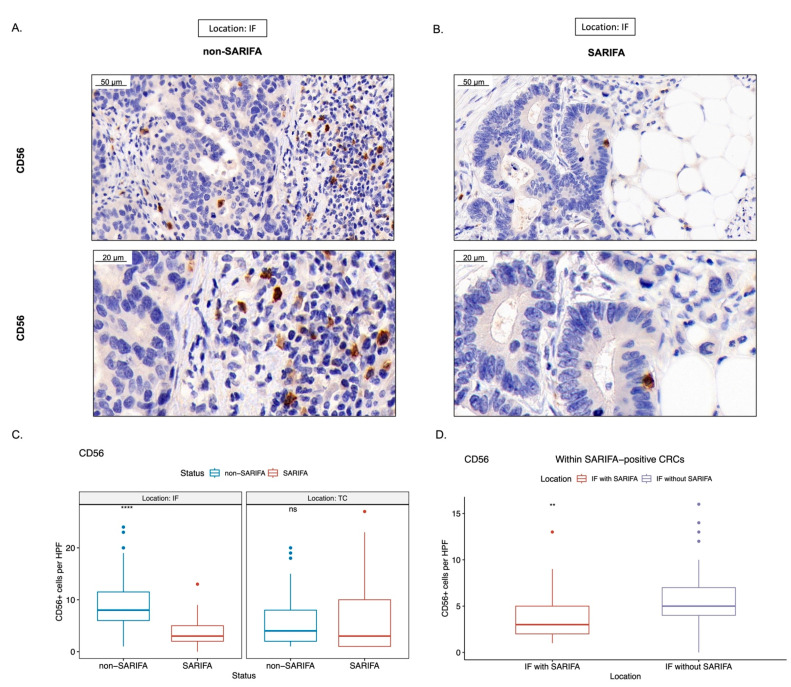
Representative immunohistochemical stains for CD56 ((**A**,**B**); scale bar = 50 µm and 20 µm, respectively). Boxplots showing the amount of CD56+ lymphocytes per high-power field (HPF, 400×), comparing between SARIFA-positive (SARIFA) and SARIFA-negative CRCs (non-SARIFA) at IF and TC (**C**) and comparing SARIFAs vs. non-SARIFAs, only within SARIFA-positive CRCs (**D**) as SARIFA-positive CRCs also show non-SARIFAs at the invasive margin (see Figure 1E). ns = not significant (*p* < 0.05). ** for *p* < 0.01, **** for *p* < 0.0001.

**Figure 5 cancers-15-00994-f005:**
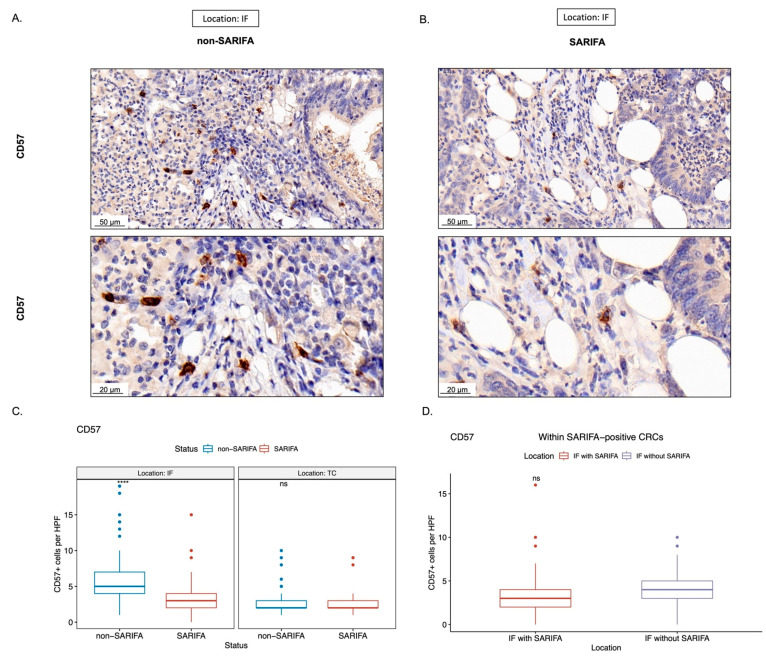
Representative immunohistochemical stains for CD57 ((**A**,**B**); scale bar = 50 µm and 20 µm, respectively). Boxplots showing the amount of CD57+ lymphocytes per high-power field (HPF, 400×), comparing between SARIFA-positive (SARIFA) and SARIFA-negative CRCs (non-SARIFA) at IF and TC (**C**), and comparing SARIFAs vs. non-SARIFAs, only within SARIFA-positive CRCs (**D**) as SARIFA-positive CRCs also show non-SARIFAs at the invasive margin (see Figure 1E). ns = not significant (at *p* < 0.05). **** for *p* < 0.0001.

**Table 1 cancers-15-00994-t001:** Demographic and Disease Characteristics of CRC patients.

Variable	SARIFA-Positive	SARIFA-Negative	p-Value
	n = 15 (33.3%)	n = 30 (67.7%)	
**Age: median (range)**	61 (42–77)	68 (48–84)	**0.043**
Gender			
male; n (%)	10 (66.7)	8 (26.7)	**0.012**
female; n (%)	5 (33.3)	22 (73.3)	
Stage			
UICC I and II; n (%)	7 (46.7)	21 (70.0)	ns
UICC III and IV; n (%)	8 (53.3)	9 (30.0)	
Tumor sidedness			
right n (%)	11 (73.3)	20 (66.7)	ns
left n (%)	4 (26.7)	10 (33.3)	
Microsatellite status			
stable n (%)	12 (80.0)	22 (73.3)	ns
unstable n (%)	3 (20.0)	7 (23.3)	
information not available		1	

ns = not significant (at *p* < 0.05). Significant *p*-values are highlighted in bold.

**Table 2 cancers-15-00994-t002:** NGS-based molecular characterization of SARIFA-positive/negative CRCs.

*Gene*	Mutational Status: wt/mut	SARIFA-Positive n = 15 *	SARIFA-Negative n = 23 *	p-Value
*KRAS*	wt	6 (40.0%)	13 (56.5%)	ns
	mut	9 (60.0%)	10 (43.5%)	
*BRAF*	wt	14 (93.3%)	18 (78.3%)	ns
	mut	1 (6.7%)	5 (21.7%)	
*NRAS*	wt	14 (93.3%)	22 (95.7%)	ns
	mut	1 (6.7%)	1 (4.3%)	
*PIK3CA*	wt	11 (73.3%)	18 (78.3%)	ns
	mut	4 (26.7%)	5 (21.7%)	
*CTNNB1*	wt	15 (100%)	22 (95.7%)	ns
	mut	0 (0%)	1 (4.3%)	
*FGFR1*	wt	15 (100%)	22 (95.7%)	ns
	mut	0 (0%)	1 (4.3%)	
*FGFR3*	wt	15 (100%)	22 (95.7%)	ns
	mut	0 (0%)	1 (4.3%)	
*ERBB2*	wt	15 (100%)	22 (95.7%)	ns
	mut	0 (0%)	1 (4.3%)	
*MET*	wt	15 (100%)	22 (95.7%)	ns
	mut	0 (0%)	1 (4.3%)	
*AKT1*	wt	15 (100%)	22 (95.7%)	ns
	mut	0 (0%)	1 (4.3%)	

ns = not significant (*p* < 0.05); mutational status: mut = mutant; wt = wildtype. * NGS-based multigene testing was possible in 38 of 45 (84.4%) cases of the entire cohort: in 15 of 15 (100%) SARIFA-positive CRCs, and in 23 of 30 (76.7%) SARIFA-positive CRCS.

**Table 3 cancers-15-00994-t003:** Lymphocyte subsets in healthy controls, SARIFA-positive and SARIFA-negative CRC patients.

	Healthy Controls n = 27	SARIFA-Positive n = 15	SARIFA-Negative n = 30	p1 SARIFApos-SARIFAneg	p2 Healthy-SARIFAneg	p3 Healthy-SARIFApos
Total lymphocytes	1795 (1195–2216)	1388 (1185–1687)	1288 (965–1623)	ns	**0.038**	ns
CD3+ cells	1101 (671–1678)	876 (783–1320)	866 (714–1142)	ns	ns	ns
CD8+ cells	228 (191–364)	228 (184–397)	233 (118–325)	ns	ns	ns
Naive	48 (9–70)	51 (33–83)	25 (15–58)	ns	ns	ns
Memory	102 (55–133)	63 (39–124)	84 (49–117)	ns	ns	ns
CM	21 (4–40)	24 (13–33)	23 (13–55)	ns	ns	ns
EM	79 (47–126)	76 (35–148)	72 (47–101)	ns	ns	ns
EMRA	68 (23–134)	69 (18–148)	29 (16–110)	ns	ns	ns
Early	160 (89–206)	133 (94–165)	78 (56–181)	ns	**0.025**	ns
Intermediate	20 (10–30)	13 (6–30)	10 (6–24)	ns	ns	ns
Late	36 (22–87)	58 (42–201)	51 (18–126)	ns	ns	ns
Exhausted	81 (51–133)	82 (29–114)	53 (23–79)	ns	**0.019**	ns
Terminal effector	21 (12–63)	39 (15–135)	31 (9–97)	ns	ns	ns
HLA-DR+	31 (10–45)	92 (56–177)	60 (36–112)	ns	**0.011**	**0.002**
CD69+	21 (17–46)	19 (12–32)	13 (7–24)	ns	**0.037**	ns
CD4+ cells	634 (487–1042)	547 (268–773)	524 (406–640)	ns	ns	ns
Naive	282 (143–371)	218 (59–441)	157 (117–239)	ns	ns	ns
Memory	411 (255–552)	311 (182–366)	303 (234–386)	ns	ns	**0.027**
CM	198 (91- 289)	190 (109–250)	190 (139–236)	ns	ns	ns
EM	181 (98–268)	99 (63–169)	123 (82–163)	ns	ns	0.05
EMRA	14 (1–44)	4 (2–8)	8 (1–26)	ns	ns	ns
Th1	28 (13–61)	17 (14–33)	19 (8–44)	ns	ns	ns
Th2	48 (34–81)	40 (16–63)	47 (33–60)	ns	ns	ns
Th17	67 (36–83)	40 (17–70)	48 (33–60)	ns	ns	ns
CD25high	11 (6–18)	10 (7–32)	15 (9–24)	ns	ns	ns
HLA-DR+	39 (27–64)	50 (34–62)	51 (44–67)	ns	**0.011**	**0.002**
CD69+	12 (7–19)	13 (11–21)	14 (9–18)	ns	**0.037**	ns
NK cells	189 (125–281)	87 (59–117)	187 (118–252)	**0.002**	ns	**<0.001**
CD56dim CD16bright	12 (9–19)	6 (4–12)	14 (8–25)	**0.004**	ns	**0.004**
CD56+ CD16+	162 (98–264)	56 (36–90)	151 (103–220)	**<0.001**	ns	**<0.001**
CD56bright CD16dim	15 (10–19)	10 (9–14)	11 (8–15)	ns	ns	ns
NK-like T cells	47 (18–83)	47 (21–67)	34 (17–134)	ns	ns	ns
B cells	208 (146–236)	122 (85–205)	123 (62–165)	ns	**<0.001**	ns
Naive	116 (88–164)	88 (58–152)	69 (37–99)	ns	**0.006**	ns
Memory	7 (5–9)	8 (4–12)	6 (2–15)	ns	ns	ns
Class switch	25 (16–41)	18 (12–21)	11 (6–28)	ns	**0.002**	ns
Transitory	7 (5–14)	1 (1–2)	2 (1–4)	ns	**<0.001**	**<0.001**
CD4/CD8 Ratio	3 (2–4)	2 (1–3)	3 (2–4)	ns	ns	ns

ns = not signficant (at *p* < 0.05). Significant *p*-values are highlighted in bold. Cell counts are given as median value/µL (interquartile range). *p*-values (adjusted) are given for pairwise comparisons using Wilcoxon test with Benjamini-Hochberg correction for following comparisons: p1) SARIFA-positive vs SARIFA-negative, p2) healthy controls vs SARIFA-negative, p3) healthy controls vs SARIFA-positive.

## Data Availability

The datasets generated during and/or analyzed during the current study are available from the corresponding author on reasonable request—and to some parts only in restricted form for privacy reasons.

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
