# Peer review of "Alterations in Natural Killer Cells in Colorectal Cancer Patients with Stroma AReactive Invasion Front Areas (SARIFA)"

_cancers, 2023, doi:10.3390/cancers15030994_

Round 1

Reviewer 1 Report

In this interesting paper authors describe a lower number of both circulating and tumor-infiltrating NK cells in a subset of CRC patients characterized by SARIFA respect to other CRC patients. However, in the present form the paper appear preliminary and simply descriptive.

A major concern is related to the lack of healthy control group. All experiments performed on PB lymphocytes may be easily repeated on healthy controls in order to appreciate the differences found between CRC groups.

Moreover, the use of CD57 in IHC is not sufficiently commented. CD56 and CD57 NK cell markers are not equivalent since CD57 label an Nk cell subpopulation that is terminally differentiated. Does this population show major differences? This marker should be added on flow cytometry analysis to understand its significance.

Functional characterization of NK cells is mandatory to understand how this lymphocyte population may correlate with poor prognosis. Degranulation potential, lytic mediator content, IFNg production may be easily performed by flow cytometry using healty controls. Moreover, activating and inhibitory receptor expression as well s the expression of checkpoint receptors is required to assess Nk cell hypofunctional state that is frequently associated to tumor progression. Some of these results may be also validated of tumor sections bu double IHC or FISH techniques in order to verify the expression of some RNA for cytokines or inhibitory receptors by tumor-infiltrating NK cells. 

I'm sure that the addition of these experiments could help to improve manuscript quality and the significance of the research.

Author Response

Dear Reviewer 1, please see the attachment, where you can find our point-point-answers as PDF file. We thank you for carefully reviewing our manuscript. In behalf of the authors, Dr. med. Nic Reitsam.

Reviewer 2 Report

This manuscript nicely describes the impact of SARIFA on the number and phenotype of NK cells in cancer patients. Methods and results are presented in a clear manner and provide the evidence for a new predictive marker in CRC.

It is very interesting the prove that SARIFA impairs the number and phenotype of NK cells. In my opinion, there are just a couple of experiments that could render this manuscript even more appealing:

1) Authors reported  NK cell subsets also showed lower values in SARIFA-positive patients for CD56dim CD16bright NK cell subsets and for CD56+ CD16+ NK cells. These observations are very interesting. CD56dim CD16bright NK cells are NK cells with the highest anti-tumor cytotoxic function, including direct NK killing and NK-mediated ADCC.

If authors have still material available could they analyze through flow cytometry if CD56dim CD16bright NK cells from SARIFA-positive patients have also an impairment in the expression of NK activation and cytotoxic markers such as NKG2D, CD107a, Perforin, Granzyme B respect to SARIFA-negative patients?

This information would be very useful to understand if SARIFA impair not only the number but also the function of cytotoxic NK cells

2) It would be very informative if authors can perform a killing assay comparing NK-killing activity of NK cells from SARIFA-positive patients with NK cells from SARIFA-negative patients using colon cancer cell lines as target.

In this way we could determine if SARIFA impair expression of  NK cell activation markers and their killing activity and this information could really makes SARIFA as a real marker to predict for example the prognosis of CRC patients treated with cetuximab, a monoclonal antibody that works through NK cells-mediated ADCC.

Author Response

Dear Reviewer 2, please see the attachment, where you will find our point-by-point answers. We thank you for your critical review and your valuable comments. On behalf of the authors, Dr. med. Nic Reitsam.

Reviewer 3 Report

The Authors proves that our newly introduced biomarker SARIFA is as-448 sociated with distinct immunologic alterations: i) a dramatic reduction in NK cells in the 449 peripheral blood of SARIFA-positive CRC patients and ii) a reduction of NK cells and NK-450 like lymphocytes at SARIFAs in CRC. However, although the association of NK cells with the course of CRC might influence CRC prognosis or treatemnt in the future, there are major drawbacks of tye study. 

1. NK cell detection in tissue samples, basing only on the CD56 experession, is inappropriate. A number of leukocyte subsets might express CD56 antigen, thus the Authors cannot conclude that CD56+ cells are NK cells.

2. The peripheral blood analysis of NK cells is limited only to their numbers and subset analysis, giving no information about the status of these cells, their activation/exhaustion or functional status.

3. The usage of different NGS panels within the study (lines 239-241) indicate the systemic bias, while the explanation how the results were is not celar.

Minor comments:

1. Inapropriate usage the term NKT cell for CD56+ T cells (DOI: 10.1016/j.jim.2017.03.016)

2. NK cell subsets are differently named in Suppl. Fig. 1 and within the text.

3. Lines 280-282 (Furthermore, SARIFA-positive patients dis-280 played approximately twice as many naïve CD8+ T cells; however, this result reached sta-281 tistical significance (50/μl vs. 23/μl; p=0.043) only slightly). It is statistically incorect to claim that p reached significance "only slightly", as if the statistical significance cutoff was set to p<0.05, all of the lower p values are significant. Thus, CD8+ role in the presented study must be disscused properly.

Author Response

Dear Reviewer 3, please see the attachment, where you will find our point-by-point answers. We thank you for critically reviewing our manuscript and all your valuable comments, which we believe improved our manuscript significantly. On behalf of the authors, Dr. med. Nic Reitsam

Round 2

Reviewer 1 Report

First, I would like to thank the authors for the addition of healthy control group, which gives the possibility to understand the defect in NK cell number in SARIFA positive patients. However, it is not clear why the old panels are still present in figure 3. A more careful revision by authors is necessary.

As already mentioned by the reviewer, the analysis of NK cell number without a functional characterization does not allow to hypothesize a different response to therapies in these patients.

I understand that blood from the analysed patients is no more available but the author can choose to recruit new patients or use Paraffin section that are available for sure.

I don't agree with the planned experiments on cultured NK cells since in vitro culture can change the activation status of these cells, I strongly recommend to perform experiments on freshly isolated NK cells from PB or by using tissue sections. Even though in a smaller cohort of patients, at least a simple functional characterization of NK cells is mandatory.

Reviewer 2 Report

Unfortunatley, authors don't have left materials to investigate phenotype of NK cells in SARIFA-positive vs SARIFA-negative patient.

I strongly encourage the authors to perform a further study to evaluate this aspect

Reviewer 3 Report

I am fine with the revision.